# HIF-1β Positively Regulates NF-κB Activity via Direct Control of TRAF6

**DOI:** 10.3390/ijms21083000

**Published:** 2020-04-24

**Authors:** Laura D’Ignazio, Dilem Shakir, Michael Batie, H. Arno Muller, Sonia Rocha

**Affiliations:** 1Centre for Gene Regulation and Expression, College of Life Sciences, University of Dundee, Dundee DD1 5EH, UK; laura.dignazio@libd.org; 2The Lieber Institute for Brain Development, Department of Neurology, Johns Hopkins School of Medicine, Baltimore, MD 21205, USA; 3Department of Biochemistry, Institute of Integrative Biology, University of Liverpool, Liverpool L69 7ZB, UK; Dilem.Shakir@liverpool.ac.uk (D.S.); M.Batie@liverpool.ac.uk (M.B.); 4Developmental Genetics Unit, Institute of Biology, University of Kassel, 34132 Kassel, Germany; uk040787@uni-kassel.de

**Keywords:** NF-κB TRAF6, HIF, ARNT, Drosophila, TNF

## Abstract

NF-κB signalling is crucial for cellular responses to inflammation but is also associated with the hypoxia response. NF-κB and hypoxia inducible factor (HIF) transcription factors possess an intense molecular crosstalk. Although it is known that HIF-1α modulates NF-κB transcriptional response, very little is understood regarding how HIF-1β contributes to NF-κB signalling. Here, we demonstrate that HIF-1β is required for full NF-κB activation in cells following canonical and non-canonical stimuli. We found that HIF-1β specifically controls TRAF6 expression in human cells but also in *Drosophila melanogaster*. HIF-1β binds to the *TRAF6* gene and controls its expression independently of HIF-1α. Furthermore, exogenous TRAF6 expression is able to rescue all of the cellular phenotypes observed in the absence of HIF-1β. These results indicate that HIF-1β is an important regulator of NF-κB with consequences for homeostasis and human disease.

## 1. Introduction

The transcription factor family NF-κB (Nuclear Factor kappa-light-chain-enhancer of activated B cells) is associated with a variety of stress responses in cells and organisms from response to pathogens to heat shock and chemotherapeutic drugs [1]. Activation of NF-κB can be achieved by three main signalling pathways, canonical, non-canonical and atypical [2], depending or not on the involvement of the Inhibitor of κB Kinase (IKK). Although some of the major components of these pathways have been identified, it is currently unclear how independent these truly are. As such, crosstalk and sharing of key molecules in these seemingly independent pathways are now being investigated.

Canonical and non-canonical NF-κB activation relies on the engagement of ligands to specific receptors on the membrane. These include the Tumour Necrosis Factor (TNF) superfamily of ligands and receptors, Toll and Interleukin-1β (IL-1β) and their receptors, as well as CD40/BAFF/LTR [1]. Upon receptor binding, recruitment of specific adaptors initiates intracellular signalling cascades. These adaptors include TNF Receptor Type 1-Associated Death Domain (TRADD), Myeloid Differentiation Primary Response gene 88 (MYD88), and TNF Receptor Associated Factors (TRAFs). TRAFs (1–6) are important in multiple steps in the NF-κB signalling cascade, and their biological relevance is exemplified by a variety of phenotypes when they are deleted in mice [3]. As such, these pathways are crucially important for immune function and the cell and organism response to inflammation [1,3].

Inflammation is also intimately connected to hypoxia, or decreased oxygen levels [4,5]. The cellular response to hypoxia is best known for the activation of the Hypoxia Inducible Factor (HIF) family of transcription factors. HIFs are heterodimers incorporating an oxygen labile HIF-α (HIF-1α, HIF-2α, HIF-3α) and a stably expressed HIF-1β (also known by its gene name *Aryl Hydrocarbon Nuclear Translocator, ARNT*) [6]. Oxygen control is achieved via a post-translational mechanism relying on the inactivation of Prolyl-Hydroxylases (PHDs) and von Hippel Lindau (VHL) tumor suppressor’s inability to ubiquitinate HIF-α in the absence of oxygen [7]. Published work demonstrated that hypoxia and inflammation have an intense molecular crosstalk in cells [4,5,8]. As such, NF-κB was shown to directly control HIF-1α, HIF-2α and HIF-1β in response to different cytokines [9,10,11,12,13,14]. Also, NF-κB interacts with HIF-1α and HIF-1β under specific conditions [15,16]. On the other hand, PHDs are reported to control NF-κB activity both in hydroxylase dependent and independent mechanisms in response to hypoxia [17], and VHL was also shown to regulate NF-κB signalling [18]. Furthermore, HIF-1α can restrict NF-κB activation in conditions of inflammation or infection [13]. In addition, and in specific immune cells, HIF-1α and HIF-2α were shown to be involved in normal immunological functions [8].

HIF-1β, although not altered in oxygen deprivation conditions, is essential for HIF activity in cells and organisms in conditions of hypoxia [6,19]. Furthermore, we had previously demonstrated that NF-κB can induce HIF-1β mRNA in response to TNF-α in human cells and in response to bacterial infection in *Drosophila melanogaster* [12,13]. Interestingly, HIF-1β was shown to bind RelB in response to CD30 stimulation of the non-canonical pathway, and control RelB transcriptional output [15]. However, whether HIF-1β is involved in broader NF-κB signalling or if this is important at the level of an organism has not been explored thus far.

In this report, we demonstrate that HIF-1β is required for full NF-κB activation in cells following canonical and non-canonical stimulation. We found that HIF-1β is required for cell survival under basal and stimulated conditions. In addition, loss-of-function of *Drosophila* HIF-1β (known as *tango, tgo*) results in reduced viability following infection. Mechanistically, we identified *TRAF6* as a gene specifically regulated by HIF-1β not only in human cancer cells but also in *Drosophila melanogaster*. Finally, exogenous expression of TRAF6 in the absence of HIF-1β was able to restore NF-κB signalling and prevent cell death. These results indicate that HIF-1β is required for full NF-κB signalling in cells, with implication in normal and disease settings.

## 2. Results

### 2.1. HIF-1β Is Required for Full NF-κB Activation in Response to Canonical and Non-Canonical Stimuli

We had previously described a role for HIF-1α in the modulation of NF-κB activation in cells and *Drosophila melanogaster* [13]. In addition, published work had indicated that HIF-1β is direct target of NF-κB [12,13], and that HIF-1β is involved in non-canonical NF-κB signalling [15]. However, whether HIF-1β is a general regulator of NF-κB or this action is restricted to CD30-mediated NF-κB activation is not known. To address this question we performed efficient siRNA mediated depletion of HIF-1β (Figure 1A, Appendix A) in multiple NF-κB-luciferase reporter cell lines and assessed activity following non-canonical (LIGHT) and canonical (TNF-α) stimulation (Figure 1B–D, Appendix A). We could determine that in response to non-canonical NF-κB stimulus, HIF-1β depletion resulted in significantly less reporter gene activity (Figure 1B). Interestingly, this was also the case when canonical signalling was initiated with TNF-α (Figure 1C,D), the opposite effect that is known for HIF-1α [13].

To determine whether our luciferase reporter results were also reflected at the level of endogenous targets, we investigated NF-κB subunits expression and some of their target genes in the presence or absence of HIF-1β following treatment with LIGHT (Figure 2) or TNF-α (Figure 3). Levels of RelB and p100 were significantly reduced in the absence of HIF-1β in response to both stimuli (Figure 2A; Figure 3A). This reduction in p100 was also observed in another cellular background, A549 lung cancer cells (Appendix A). In response to LIGHT, we could observe significant reduction in the mRNA levels of p100 and RelB, as well as in Rantes, a target of the non-canonical NF-κB pathway (Figure 2B), when HIF-1β was depleted. Furthermore, other p52-dependent target genes [20,21] were also reduced at protein level in the absence of HIF-1β (Figure 2C).

In response to TNF-α, p105 levels were also reduced (Figure 3A) in the absence of HIF-1β, a predicted result since these NF-κB subunits are under the direct control of RelA [22]. Also, as expected, and given that it is not an NF-κB target, RelA levels were unaffected by HIF-1β depletion in response to TNF-α (Figure 3A). Importantly, NF-κB targets such as p100 and IL-8 mRNA levels were also reduced in the absence of HIF-1β following TNF-α treatment (Figure 3B,C). Furthermore, when we overexpressed HIF-1β, levels of p100 and IL-8 mRNA were also significant increased following TNF-α treatment (Appendix A), suggesting that indeed HIF-1β levels have an impact on NF-κB signaling in the cells we investigated.

It is well known that NF-κB signalling in response to TNF-α is used to prevent cell death [23]. To understand the biological impact of HIF-1β, we investigated cellular apoptosis markers as well as organism survival using the model system *Drosophila melanogaster* (Figure 4). Our analysis revealed that HIF-1β depletion led to increased levels of apoptotic markers in control, LIGHT and TNF-α treated cells (Figure 4A,B). Increased levels of cleaved Poly-ADP Ribose Polymerase (PARP), cleaved caspase-3 and cleaved caspase-7 were observed when HIF-1β levels were reduced by siRNA (Figure 4A,B). In *Drosophila*, NF-κB mediates innate immunity responses to infection [24]. We had previously demonstrated that loss of NF-κB in *Drosophila* results in reduced viability in response to infection with bacterial strains such as *Serratia marcescens* [13]. Importantly, at the level of the whole organism, loss of function of *Drosophila* HIF-1β, Tango (Appendix A), resulted in reduced viability in response to infection with *Serratia marcescens* (Figure 4C).

These results are intriguing, as deletion of *Drosophila* HIF-1α, Sima, produces similar results but due to over active NF-κB [13]. As such, we analysed levels of an NF-κB target, Drosomycin, as well as one of the NF-κB subunits in *Drosophila*, Dorsal, with or without Tango in response to infection with *E. coli* (Figure 4D). While the absence of Tango resulted in reduced levels of Drosomycin, we observed increases in the levels of Dorsal (Figure 4D) and Attacin A (Appendix A), in a manner analogous to loss of Sima [13]. These results suggest that deletion of Tango in *Drosophila* impacts on NF-κB activity in mechanisms that are dependent and independent of its partner Sima. Taken together, these findings indicate that HIF-1β is required for an appropriate NF-κB signalling in cells and *Drosophila melanogaster*.

### 2.2. HIF-1β Is Required for TRAF6 Expression

To investigate the mechanism behind HIF-1β involvement in NF-κB signalling, we next analysed levels of IKK activation markers such as phosphorylated IKK and IκB-α (Figure 5A). In the absence of HIF-1β, we observed reduced levels of basal IκB-α, as expected, due to it being a NF-κB target. Furthermore, levels of p-IKK and p-IκB-α were also reduced. Similarly, following LIGHT stimulation, level of NF-κB inducing Kinase (NIK) were reduced (Figure 5B). These results suggested that HIF-1β is regulating a component upstream of the IKK complex. Directly upstream of IKK, are the Transforming Growth Factor (TGF)-β Kinase (TAK), and TAK Binding Protein (TAB) complex [25]. We thus analysed levels of the TAK-TAB complex, TRAFs but also Inhibitor of Apoptosis 1 (IAP1) and Receptor Interacting Serine/Threonine Kinase 1 (RIP1) (Figure 5C), proteins involved in different parts of the TNF-α signalling cascade to NF-κB [1]. Unexpectedly, we observed increases in the protein levels of IAP1, TRAF1 and TRAF5 (Figure 5C). No changes were consistently seen for RIP1, TAK1 and TAB1 (Figure 5C, Appendix A). However, we observed reduced levels of TRAF6 in both cellular backgrounds we investigated (HeLa and A549), in the presence or absence of TNF-α, when HIF-1β was depleted (Figure 5C, Appendix A). The reduction of TRAF6 was also seen at transcriptional level when HIF-1β was depleted (Figure 3C), but the opposite was observed for TRAF3 (Appendix A), suggesting a pleotropic effect for HIF-1β. Importantly, loss of function for HIF-1β in *Drosophila*, also resulted in reduced levels of *Drosophila* TRAF6 (dTRAF6) mRNA (Figure 5E), implying that this is a conserved function for HIF-1β across organisms.

### 2.3. HIF-1β Binds to the TRAF6 Promoter and Controls TRAF6 Expression Independently of HIF-1α

Since HIF-1β is part of several transcription factor complexes including HIF and Aryl Hydrocarbon Receptor (AHR), we interrogated the publicly available datasets for reports of HIF-1β binding at genomic areas controlling the *TRAF6* gene. Our analysis revealed that in the breast cancer cell line T47D, HIF-1β binds at the proximal promoter of *TRAF6*, 178 bp upstream to 366 bp downstream of the transcription start site (TSS) (TSS -178/+366) (Figure 6A) [26]. In combination with the analysis of other ENCODE datasets, we also performed a bioinformatic analysis for potential HIF-1β binding sites (HREs and AHR) in *TRAF6* (Appendix A) using the ALGGEN PROMO software tool. This revealed several potential hypoxia response elements (HRE) sites and AHR binding sites located both upstream and downstream to *TRAF6* TSS (Appendix A). We thus investigated if in our cell model, the site identified in T47D dataset and the AHR sites identified by our bioinformatic analysis were *bona fide* sites occupied by HIF-1β by Chromatin ImmunoPrecipitation (ChIP)-qPCR (Figure 6B). Our analysis revealed that both the TSS -178/+366 site and a putative AHR site mapping 524 bp downstream of *TRAF6* TSS (TSS +524) were significantly enriched for HIF-1β above control antibody levels (Figure 6B), while control regions of *TRAF6* genes did not contain any HIF-1β binding (Appendix A). Interestingly, and also as predicted by our western blot analysis, treatment with TNF-α did not change HIF-1β levels present at the *TRAF6* gene in either of the regions we analysed (Figure 6B).

As mentioned above, HIF-1β is a known component of several transcription factor complexes [27]. As we had previously implicated HIF-1α in the control of NF-κB signalling [13], we determined if HIF-1α was equally important for the control of TRAF6 levels. Western blot analysis indicated that HIF-1α depletion results in slightly increased levels of TRAF6 (Figure 6C), the opposite results obtained following HIF-1β siRNA-mediated knockdown. These findings suggest that HIF-1α plays an inhibitory role for TRAF6 expression. Interestingly, although ChIP-qPCR analysis revealed a significant level of HIF-1α binding in control conditions to the TSS -178/+366 site of the *TRAF6* gene, treatment with TNF-α resulted in a significant decrease of HIF-1α binding to this site (Figure 6D). As expected, control regions of *TRAF6* had no significant HIF-1α binding (Appendix A). These data suggest that HIF-1β control of *TRAF6* gene occurs independently of HIF-1α. We also investigated the potential involvement of HIF-2α (Figure 6E; Appendix A) and AHR (Figure 6F; Appendix A) in the control of TRAF6 expression. As observed with HIF-1α, HIF-2α depletion results in increased levels of TRAF6. However, depletion of AHR, results in reduced TRAF6 protein levels but not mRNA (Figure 6F; Appendix A). These results indicate that HIF-2α and AHR are not involved in TRAF6 control mediated by HIF-1β. In particular, AHR alters TRAF6 protein in a manner independent of transcriptional regulation.

### 2.4. Exogenous TRAF6 Rescues NF-κB Signalling Defect in Cells Depleted of HIF-1β

Our analysis revealed that TRAF6 is directly regulated by HIF-1β, with a potential impact on NF-κB signalling. Therefore, we hypothesised that if TRAF6 is important for HIF-1β control over NF-κB, restoration of TRAF6 would rescue the phenotypes we observed when HIF-1β was depleted. As such, we started this analysis using our reporter cell line. Here, exogenous TRAF6 was able to fully rescue NF-κB luciferase activity in the absence of HIF-1β (Appendix A) following TNF-α stimulation. Levels of endogenous p100 were also fully restored when TRAF6 overexpression was combined with HIF-1β depletion upon treatment with TNF-α or LIGHT (Figure 7A,B). We also investigated IKK and IκB-α phosphorylation and levels. Here, overexpression of TRAF6 partially rescued levels of IKK phosphorylation but completely rescued levels of p-IκB-α (Figure 7C). These findings strongly suggest that, indeed, TRAF6 regulation by HIF-1β significantly impacts NF-κB signalling following TNF-α.

Given that we had found that HIF-1β depletion resulted in increased levels of apoptotic markers in cells, we also investigated the importance of TRAF6 in this phenotype. Exogenous expression of TRAF6 strongly reduced the increase in apoptotic markers in cells depleted of HIF-1β (Figure 7D). This was observed in diminished levels of cleaved PARP, cleaved caspase-3 and cleaved caspase-7. Taken together, these results imply that HIF-1β-mediated control of TRAF6 is required for full NF-κB activity in cells, and, as such, this regulatory mechanism controls cell survival.

## 3. Discussion

In this report, we identified that HIF-1β, via the control of TRAF6, is required for full activation of NF-κB signalling in human cancer cell models. The functional importance of HIF-1β for controlling TRAF6 levels, and survival, was also observed in the model organism *Drosophila melanogaster*. Our previous published work indicated that loss of the HIF-1α homologue in *Drosophila*, Sima, also resulted in reduced viability of flies when infected with bacteria [13]. Mechanistically, this was due to elevated levels of NF-κB dependent target genes. Here, we show that HIF-1β depletion results in reduced levels of NF-κB activation in cells. In *Drosophila*, depletion of Tango (HIF-1β homologue) results in both reduced and increased levels of NF-κB targets. This implies that Tango shares some overlapping functions in response to infection with Sima, but also suggests that it possesses Sima independent functions.

TRAF6 is an adaptor protein able to bridge signalling pathways derived from TNFR and IL-1β/Toll Receptors (reviewed in [28]). It possesses a TRAF domain, used for protein-protein interaction and a ring ubiquitin ligase domain, needed for poly-ubiquitination of substrates as well as auto-ubiquitination [3]. While the absolute requirement for the TRAF domain is well established, the requirement for the ubiquitinase function of the protein for signalling is controversial and context dependent [29]. TRAF6 knockout mice are prenatal and postnatal lethal [30]. Conditional knockout models revealed essential roles for TRAF6 in a variety of immune cellular backgrounds but also in epithelial cell types [31,32,33,34], indicating how important this factor is for normal cellular homeostasis. More recently, TRAF6 was also implicated in cancer signalling, by regulating epithelial to mesenchymal transition (EMT) in colon cancer [35], DNA damage in breast cancer backgrounds [36], and involvement in autophagy responses [37]. Our results demonstrate the importance of TRAF6 in cell survival, following NF-κB stimulation, in cancer cells. This is in line with reduced NF-κB transcriptional activity required for the activation of pro-survival genes following TNF-α [23]. Interestingly, inhibition of TRAF6 was also associated with increased apoptosis when cells were treated with cinchonine, a natural occurring chemical compound [38].

Although much is known about TRAF6 regulation at the post-translational level, very little is known about how its promoter is controlled. TRAF6 mRNA was found to be over expressed in cancer [39], but the mechanisms underlying this increased expression have not been determined yet. Our data identified *TRAF6* as a gene specifically down-regulated when HIF-1β, and its homologue in *Drosophila melanogaster*, Tango, are depleted. Analysis, using Ominer [40,41], of public datasets for ChIP-seq revealed that several transcription factors occupy genomic regions within the TRAF6 promoter, including Basic Helix-Loop-Helix ones such as HIFs. Furthermore, analysis of publicly released datasets for HIF-1β ChIP-seq identified a specific binding site in the promoter region of TRAF6 in T47D breast cancer cells [26]. We could validate this occurrence in the cell lines used in this study, and we also identified another binding site in the vicinity of *TRAF6* transcriptional start site. These results suggest a potential direct control of HIF-1β over the *TRAF6* gene. Interestingly, although we could detect HIF-1α binding to one of these sites, TRAF6 expression increased following depletion of HIF-1α, suggesting that HIF-1β control over TRAF6 occurs independently of HIF, potentially via another of its binding partners. AHR depletion, while reducing TRAF6 protein did not reduce TRAF6 mRNA levels, suggesting that AHR controls TRAF6 translation or protein stability indirectly. Another possibility would be that HIF-1β impacts on NF-κB-dependent regulation of TRAF6. In fact, NF-κB also possesses several binding sites in the *TRAF6* promoter, and given that we found that HIF-1β depletion significantly reduces NF-κB transcriptional activity, this would be a potential mechanism behind TRAF6 mRNA decrease. Analysis of RelA ChIP-seq datasets available on ENCODE [42,43], indicated that RelA binds to one of the sites we identified as HIF-1β binding site to the *TRAF6* gene (TSS -178/+366). It is possible that a cooperative binding between HIF-1β and RelA occurs. Future studies depleting NF-κB could help answer these questions.

The implications of our findings are relevant for situations of homeostatic control but also pathological deregulation such as inflammation and cancer. It would be predicted that a strong correlation between HIF-1β and TRAF6 would occur in tissues both in normal conditions as well as in disease situations. Analysis of TGCA data and expression correlation analysis using GEPIA 2 [44] revealed that HIF-1β and TRAF6 mRNA are very strongly correlated in normal tissues such as pancreas and breast (Appendix A). In cancer situations, this correlation in mRNA levels was found to be also strong in pancreatic cancer, breast cancer, kidney renal cell carcinoma (RCC), uveal melanoma, and diffuse B-cell lymphoma (Appendix A). These analyses are remarkably supportive of our cellular findings.

TRAF6 function in *Drosophila melanogaster* was also shown to connect NF-κB and survival pathways [45,46]. In, addition, dTRAF6 was shown to control the Notch signalling in these organisms [47]. Our results demonstrated that dHIF-1β is required for dTRAF6 levels, and its loss of function also resulted in reduced viability in response to infection, suggesting a conservation of the responses we observed in cancer cells. It would be interesting to determine if the *dTRAF6* promoter also contains dHIF-1β binding sites. Further studies are needed also to establish a direct regulatory mechanisms in *Drosophila*.

Taken together our analyses revealed an unexpected link between HIF-1β and NF-κB signalling, suggesting that these transcription factors are strongly linked in signal transduction in homeostasis and disease.

## 4. Materials and Methods

### 4.1. Cell Lines and Growth Conditions

All cells, with the exception of 786-O, were maintained in Dulbecco’s modified Eagle’s medium (DMEM, Lonza, Slough, UK) supplemented with 10% fetal bovine serum (FBS, Invitrogen/ThermoFisher, Paisley, UK), 1% penicillin-streptomycin (Lonza, Slough, UK), and 1% L-glutamine (Lonza, Slough, UK) at 5% CO_2_ and 37 °C for no more than 30 passages. 786-O cells were maintained in RPMI (Gibco/ThermoFisher, Paisley, UK), supplemented as above. Cells were routinely tested for mycoplasma contamination using MycoAlert Mycoplasma Detection Kit (Lonza, Slough, UK).

Human cervix carcinoma HeLa, and human lung carcinoma A549 were obtained from the American Type Culture Collection (ATCC, Manassas, VA, USA).

HeLa-κB Luciferase cells, containing an integrated copy of the 3x-κB ConA luciferase reporter plasmid and described in [48], were kindly gifted by Prof. Ron Hay (School of Life Sciences, University of Dundee, UK). A549-κB Luciferase cells were previously created in the lab, by transfecting the NF-κB-reporter construct pGL4.32(luc2P/NF-κB-RE/Hygro), selecting individual clones on the basis of their response to TNF-α, and maintaining the derived cell lines with 150 μg/mL Hygromycin B (Sigma, Gillinham, UK) [49].

### 4.2. Cell Transfection

Small interfering RNA oligonucleotides were purchased from Eurofins Genomics (Ebersberg, Germany) and used in a final concentration of 27 nM. siRNA transfections in all cell lines were performed using Interferin (Polyplus, Illkirch, France) according to manufacturer’s instructions. Oligonucleotide sequences used for siRNA knockdown are as follows: Control- AACAGUCGCGUUUGCGACU; HIF-1β_#1- GGUCAGCAGUCUUCCAUGA; HIF-1β_#2- GAAAGAAACAUGUGAGUAA; HIF-1α - GCAUAUAUCUAGAAGGUAU; HIF-2α_#1- CAGCAUCUUUGAUAGCAGUTT; HIF-2α _#2- GGCAGAACUUGAAGGGUUA; AHR_#1- UACUUCCACCUCAGUUGGCTT; AHR_#2- GGACAAACUUUCAGUUCUU.

To perform DNA plasmid overexpression in all cell lines, TurboFect (ThermoFisher, Paisley, UK) or GeneJuice (Novagen/ThermoFisher, Paisley, UK) were used according to manufacturer’s instructions. The following expression plasmids were used in this study: GFP-C3 (Clontech/Takara, Montain View, CA, USA); Flag-pcDNA3.1 (a gift from Stephen Smale, Addgene plasmid #20011, Watertown, MD, USA); pEBB-HIF-1β−GFP (kind gift from Colin Duckett, Ann Habour, MI, USA); pCMV5-FLAG TRAF6 (MRC-PPU reagents, Dundee, UK).

### 4.3. Cell Treatments

To stimulate the inflammatory response, human recombinant TNF-α (Peprotech, London, UK) and human recombinant LIGHT (TNFSF14, Peprotech, London, UK) were dissolved in sterile PBS and used at a final concentration of 10 ng/mL and 100 ng/mL, respectively.

### 4.4. Luciferase Assay

2 × 10^5^ cells stably transfected with a luciferase reporter gene were seeded in 6-well plates, and transfected with appropriate siRNA or DNA plasmid, according to procedures previously described above. Then, cells were stimulated with TNF-α or LIGHT (κB luciferase reporter), for the indicated times, and harvested with 400 µL of Passive Lysis Buffer (Promega, Southampton, UK). Luciferase activity was measured according to manufacturer’s instructions (Luciferase Assay System, Promega, Southampton, UK), and normalised to protein concentration (Bradford, BioRad, Watford, UK). All experiments were performed a minimum of three times before calculating means and standard error of the means.

### 4.5. RNA Extraction cDNA Synthesis and Real Time Quantitative PCR Analysis

PeqGOLD total RNA kit (Peqlab, Bishop’s Waltham, UK) or PureLink RNA Mini Kit (Life Technologies/ThermoFisher, Paisley, UK) were used to extract total RNA from cells according to the manufacturer’s instructions. RNA was converted to cDNA using Quantitect Reverse Transcription Kit (Qiagen, Manchester, UK) or First Strand cDNA Synthesis kit (ThermoFisher, Paisley, UK). For quantitative PCR, Brilliant II Sybr green kit (Stratagene/Agilent, Stockport, UK), and recommended MX3005P 96-well skirted plates were used to analyse samples on the Mx3005P qPCR platform (Stratagene/Agilent, Stockport, UK). Alternatively, PerfeCTa Sybr Green FastMix (Quanta Bio, Bervely, MA, USA) with ROX dye added in 1:250 ratio, and recommended Microamp Optical 96-well reaction plates were used to analyse samples on the QuantStudio 6 Flex qPCR platform (Applied Biosystem/ThermoFisher, Paisley, UK). Actin or 18S were used as normalising genes. RT-PCR results were analysed by the ∆∆Ct method. The primers used for gene expression analysis by RT-PCR are: 18S, For: 5′-AAACGGCTACCACATCCAAG-3′, Rev: 5′-CGCTCCCAAGATCCAACTAC-3′. Actin, For: 5′-CTGGGAGTGGGTGGAGGC-3′, Rev: 5′-TCAACTGGTCTCAAGTCAGTG-3′. HIF-1β, For: 5′-CAAGCCCCTTGAGAAGTCAG-3′, Rev: 5′-GAGGGGCTAGGCCACTATTC-3′. IL-8, For: 5′-CCAGGAAGAAACCACCGGA-3′, Rev: 5′-GAAATCAGGAAGGCTGCCAAG-3′. p100, For: 5′-AGCCTGGTAGACACGTACCG-3′, Rev: 5′-CCGTACGCACTGTCTTCCTT-3′. TRAF3, For: 5′-CTCACAAGTGCAGCGTCCAG-3′, Rev: 5′-GCTCCACTCCTTCAGCAGGTT-3′. TRAF6, For: 5′-CCTTTGGCAAATGTCATCTGTG-3′, Rev: 5′-CTCTGCATCTTTTCATGGCAAC-3′. AHR, For: 5′-ACTCCACTTCAGCCACCATC-3′, Rev: 5′-ATGGGACTCGGCACAATAAA-3′. Rantes, For: 5′-GTCGTCTTTGTCACCCGAAAG-3′, Rev: 5′-TCCCGAACCCATTTCTTCTCT-3′. RelB, For: 5′- TCCCAACCAGGATGTCTAGC-3′, Rev: 5′-AGCCATGTCCCTTTTCCTCT-3′.

### 4.6. Protein Lysis and Western Blotting

Cells were lysed using 100 µL of whole cell protein lysis buffer (20 mM Tris pH 7.6, 150 mM NaCl, 0.75% NP-40, 5 mM NaF, 500 µM Na_3_VO_4_, and 1 Pierce Protease Inhibitor Mini Tablet (EDTA Free, ThermoFisher, Paisley, UK) per 10 mL of buffer) or RIPA buffer (50 mM Tris pH 8, 150 mM NaCl, 1% NP-40, 0.5% sodium deoxycholate, 0.1% SDS, 10 mM NaF, 1 mM Na_3_VO_4_, and 1 Pierce Protease Inhibitor Mini Tablet (EDTA Free, ThermoFisher, Paisley, UK) per 10 mL of buffer). Upon collection, cells were kept on ice for 30 min before centrifugation at 16,060× *g* at 4 °C for 15 min. The supernatant was collected and stored at −80 °C. Protein concentration was determined using Bradford (BioRad, Watford, UK) method. 20–30 µg of protein was prepared in 2× SDS loading buffer (100 mM Tris-HCl pH 6.8, 20% glycerol, 4% SDS, 200 mM DTT, and Bromophenol Blue), and incubated for 5–10 min at 105 °C. Western blotting was performed as described in [12]. Briefly, samples were loaded into an SDS-page gel (Tris-HCl poly-acrylamide gel) previously prepared and run at 80–120 volts in Running Buffer (25 mM Tris, 0.195 M glycine, and 0.1% SDS). The gel was then transferred in a semi-dry transfer (BioRad, Watford, UK) into a PVDF membrane (Millipore, Feltham, UK) for 1.5–2 h at 15 volts/0.80 mA in Transfer Buffer (50 mM Tris, 40 mM glycine, 0.001% SDS and 10% methanol). Then, the membrane was blocked with 10% Milk in TBS-tween buffer (20 mM Tris pH 7.6, 150 mM NaCl, 0.1% Tween) for 10 min, followed by three 5 min washes with TBS-tween buffer. Membranes were incubated with primary antibodies for 1 h at room temperature or overnight at 4 °C, in accordance with primary antibodies’ manufacturer instructions. Primary antibodies purchased from Cell signalling (Leiden, The Netherlands) were: caspase 3 (#9665S), caspase 7 (#128275), cleaved caspase 3 (#9664S), cleaved caspase 7 (#8438S), cleaved PARP (#5625), HIF-1β (#5537S), IAP1 (#4952), IKK-α (#2682), IκB-α (#4821), PARP (#9532S), phospho-IKK-α (#2681S), phospho-IκB-α (#9246S), RelB (#10544), RIP (#4926), TAB1 (#3225), TAK1 (#4505), TRAF1 (#4715), TRAF2 (#4724), TRAF5 (#41658), TRAF6 (#8028), HIF-2α (#7096), SKP2 (#4358), NIK (#4994). Primary antibodies against HIF-1α (#610958) were from BD Biosciences (Workingham, UK), anti-p100/p52 (#05-361) from Millipore (Feltham, UK), Cyclin B1 (#ab137875) from Abcam (Cambridge, UK), p105/p50 (#sc-7178) and RelA (#sc-372) from Santa Cruz (Dallas, TX, USA), and anti-b-Actin (#66009-1-1g) from Proteintech (Manchester, UK). Membranes were then washed three times with TBS-tween and incubated with the appropriate secondary HRP antibody (anti-mouse IgG, HRP-linked, Cell Signalling #7076, Leiden, Holland; anti-rabbit IgG, HRP-linked, Cell Signalling #7074, Leiden, The Netherlands). After washes, membranes were developed using ECL solution (Pierce/ThermoFisher, Paisley, UK).

### 4.7. Chromatin Immunoprecipitation

Chromatin Immunoprecipitation (ChIP) was performed adapting the protocol described in [21]. Cells were plated and grown to 70–80% confluency on 150 mm plates in 16 mL of appropriate culturing media. When indicated cells were treated with TNF-α for 4 h. Then, proteins and chromatin were cross-linked with 1% formaldehyde at 37 °C for 10 min. To quench the cross-linking, glycine was added to a final concentration of 0.125 M for 5 min at 37 °C. Cells were washed twice with ice-cold PBS, then scraped and centrifuged at 1000 rpm in a Beckman Coulter’s Allegra X-12 benchtop centrifuge for 5 min. The supernatant was removed and the pellet resuspended in 400 μL of ChIP lysis buffer (1% SDS, 10 mM EDTA, 50 mM Tris-HCl pH 8.1, and 1 Pierce Protease Inhibitor Mini Tablet (EDTA Free, ThermoFisher, Paisley, UK), up to 10 mL of buffer), before being snap-frozen in dry ice and stored at −80 °C. Once thawed on ice, 200 µL aliquots of each sample were transferred into 1.5 mL TPX Polymethylpentene (PMP) tubes (Diagenode, Seraing, Belgium) to improve sonication and shearing efficiency. Samples were sonicated in Bioruptor NGS (Diagenode, Seraing, Belgium) at 4 °C for five cycles of 30 s ON/30 s OFF, at high intensity amplitude. This sonication procedure was repeated four times. Supernatants were recovered by centrifugation at maximum speed in a benchtop centrifuge for 10 min at 4 °C prior storage of 10% of each sample as input. Remaining samples were split into 120 μL aliquots before being diluted 10 fold in ChIP Dilution Buffer (1% Triton X-100, 2 mM EDTA, 150 mM NaCl, 20 mM Tris-HCl pH 8.1). Diluted samples were pre-cleared for 2 h at 4 °C by incubation with 2 μg of sheared salmon sperm DNA and 20 μL of protein G-Sepharose (50% slurry), previously washed in cold PBS. Immunoprecipitation was performed overnight on the remaining samples with addition of appropriate antibodies (5 µL of anti-HIF-1α (Active Motif, #39665, La Hupe, Belgium), 4 µg of anti-HIF-1β (Bethyl, #A302-765A Montegomery, TX, USA) or Normal Rabbit IgG (#I5381, Sigma, Gillinham, UK) as negative control, and 0.1% of BRIJ-35 detergent. The following day, immune complexes were captured by incubation with 30 μL of protein G-Sepharose (50% slurry, previously washed with cold PBS) and 2 μg of salmon sperm DNA for 2.5 h at 4 °C. The immunoprecipitates were washed sequentially for 5 min at 4 °C in 1 mL of cold Wash Buffer 1 (0.1% SDS, 1% Triton X-100, 2 mM EDTA, 20 mM Tris-HCl pH 8.1, 150 mM NaCl), 1mL of cold Wash Buffer 2 (0.1% SDS, 1% Triton X-100, 2 mM EDTA, 20 mM Tris-HCl pH 8.1, 500 mM NaCl), and 1 mL of cold Wash Buffer 3 (0.25 M LiCl, 1% Nonidet P-40, 1% deoxycholate, 1 mM EDTA, 10 mM Tris-HCl pH 8.1). Next, beads were washed twice with 500 µL of Tris-EDTA (TE) buffer.

Chromatin reverse cross-linking and DNA elution were performed using the IPure kit v2 (Diagenode, Seraing, Belgiumfollowing manufacturer’s instructions. Briefly, 10% inputs and beads were resuspended in 90 µL and 100 µL of Elution Buffer mix, respectively. The elution buffer mix was prepared with 115.4 µL of Buffer A and 4.6 µL of Buffer B per sample. All samples were incubated overnight at 65 °C on a thermomixer with continuous shaking at 300 rpm. The following day, supernatants were recovered by centrifugation at 1000 rpm for 1 min in a benchtop centrifuge. 2 µL of carrier, 100 µL of 100% isopropanol and 10 µL of magnetic beads were added to each input and immunoprecipitate sample, then incubated on a rotating wheel (40 rpm) for 10 min at room temperature. After a quick centrifugation, tubes were placed into a magnetic rack for separation of buffer, then discarded, and beads. Captured beads were gently mixed to 100 µL of Wash Buffer 1 (previously diluted with 100% isopropanol with 1:1 ratio), prior incubation on rotating wheel (40 rpm) for 5 min at room temperature. Following another step of buffer separation, beads were gently mixed to 100 µL of Wash Buffer 2 (previously diluted with 100% isopropanol with 1:1 ratio), and incubated on rotating wheel (40 rpm) for 5 min at room temperature. Finally, DNA elution was performed adding to captured beads 25 µL of Buffer C and incubating tubes on a rotating wheel (40 rpm) for 15 min at room temperature. After separation on magnetic rack, the first fraction of eluted DNA was transferred into a new storage tube, while captured beads were subjected to a second step of DNA elution by adding other 25 µL of Buffer C, to obtain a final volume of 50 µL of purified DNA.

3 μL DNA were used for RT-PCR analysis with primers specifically designed on TRAF6 promoter regions: TRAF6 (TSS -178/+366), For: 5′-AAGGAGACTCACCGTTCTA-3′, Rev: 5′-TCTGTGTCCGTCCTCTAC-3′. TRAF6 (TSS +524), For: 5′-GGTCGAGGACACCGTTC-3′, Rev: 5′-GTGGAATGAGCGAGGAAGA-3′. When performed quantitative PCR on ChIP samples, primers designed on coding regions were used as negative control.

### 4.8. Chromatin Immunoprecipitation Sequencing Analysis

T47D cell HIF-1β ChIP-seq data was downloaded from the NCBI SRA (SRX666557) and GEO (GSM1462476). Coverage tracks were generated using the R Bioconductor package, Gviz [50]. RelA ChIP-seq datasets were downloaded from the ENCODE portal [46] (https://www.encodeproject.org/) with the following identifiers: ENCSR000EBM/ENCFF002CQN, ENCSR000EAG/ENCFF002CPA, ENCSR000EAN/ENCFF002CQB, ENCSR000EBD/ENCFF002CQJ, ENCSR772EEN/ENCFF580QGA.

### 4.9. Statistical Analysis

Means, standard deviations and standard error means were calculated on a minimum of three independent biological experiments. Student *t* tests was used when comparing two conditions only; one way ANOVA analysis followed by Dunnett test was used for comparing multiple conditions to a control condition only; one way ANOVA analysis followed by Tukey test was used for multiple pair-wise condition comparisons. In all cases *p*-values were calculated as follows: * *p* < 0.050; ** *p* < 0.010 and *** *p* < 0.001.

### 4.10. Drosophila Melanogaster

Fly culture and husbandry were performed according to standard protocols. The *tgo* allele P{EPgy2}*tgo*^EY03802^ was used in all experiments. represents a P-Element insertion within the first exon of the *tgo* gene. For the experiments, we used heterozygous *tgo*^EY03802^ flies, which exhibit reduced levels of *tgo* mRNA. *white*^1118^ flies were used as control. For any details about fruit flies used in this study refer to the content on FlyBase website (http://flybase.org/).

For RNA extraction and RT-PCR analysis, ~70 adult flies were placed in a 1.5 mL tube, frozen in liquid nitrogen and mechanically disrupted by vortexing. *Drosophila* heads were collected and homogenised with 250 µL of appropriate Lysis Buffer. RNA extraction was performed using PeqGOLD total RNA kit (Peqlab, Bishop’s Waltham, UK) or PureLink RNA Mini Kit (Life Technologies/ThermoFisher, Paisley, UK) according to manufacturer’s instructions.

RNA was converted to cDNA using Quantitect Reverse Transcription Kit (Qiagen, Manchester, UK) or First Strand cDNA Synthesis kit (ThermoFisher, Paisley, UK). For quantitative PCR, Brilliant II Sybr green kit (Stratagene/Agilent, Stockport, UK), and recommended MX3005P 96-well skirted plates were used to analyse samples on the Mx3005P qPCR platform (Stratagene/Agilent, Stockport, UK). Alternatively, PerfeCTa Sybr Green FastMix (Quanta Bio) with ROX dye added in 1:250 ratio, and recommended Microamp Optical 96-well reaction plates were used to analyse samples on the QuantStudio 6 Flex qPCR platform (Applied Biosystem/ThermoFisher, Paisley, UK). dActin was used as normalising gene in all experiments, and RT-PCR results were analysed by the ∆∆Ct method. Primers used for qPCR are: dActin, For: 5′-GCGTTTTGTACAATTCGTCAGCAACC-3′, Rev: 5′-GCACGCGAAACTGCAGCCAA-3′. dTraf6, For: 5′-TCAGAACCATCACTATGCACCC-3′, Rev: 5′-CAAATGGCGCACTCGTATCG-3. Tgo, For: 5′-CGGCTGCTCATACGCCCGAG-3′, Rev: 5′-GGCCAGCATGTGCGTCTGGT-3′. Dorsal, For: 5′-TGTTCAAATCGCGGGCGTCGA-3′, Rev: 5′-TCGGACACCTTCGAGCTCCAGAA-3′. Drosomycin, For: 5′-GTTCGCCCTCTTCGCTGTCCTGA-3′, Rev: 5′-CCTCCTCCTTGCACACACGACG-3′. Attacin A, For: 5′-AGGTTCCTTAACCTCCAATC-3′, Rev: 5′-CATGACCAGCATTGTTGTAG-3′.

Protein concentration was determined using Pierce BCA Protein Assay Kit (ThermoFisher, Paisley, UK) according to the manufacturer’s instructions. 20 μg of protein were prepared in 2× SDS loading buffer and processed for Western Blot analysis as described in the previous section. Primary antibody against Tango was purchased from DSHB (Iowa City, IA, USA).

To measure survival rate upon bacterial infection, adult flies were pricked with a Tungsten needle inoculated with a concentrated bacterial solution (*Serratia marcescens* (OD600 < 0.20)) and incubated for 24 h (Survival analysis) at 25 °C.

In general, means, standard deviations and standard error means were calculated prior performing Student *t* tests on a minimum of three independent biological experiments and calculating *p*-values. * *p* < 0.050; ** *p* < 0.010 and *** *p* < 0.001.

For *Drosophila melanogaster* survival studies, *p*-values were obtained using Log-Rank statistical analysis. * *p* < 0.050; ** *p* < 0.010 and *** *p* < 0.001.

## Figures and Tables

**Figure 1 ijms-21-03000-f001:**
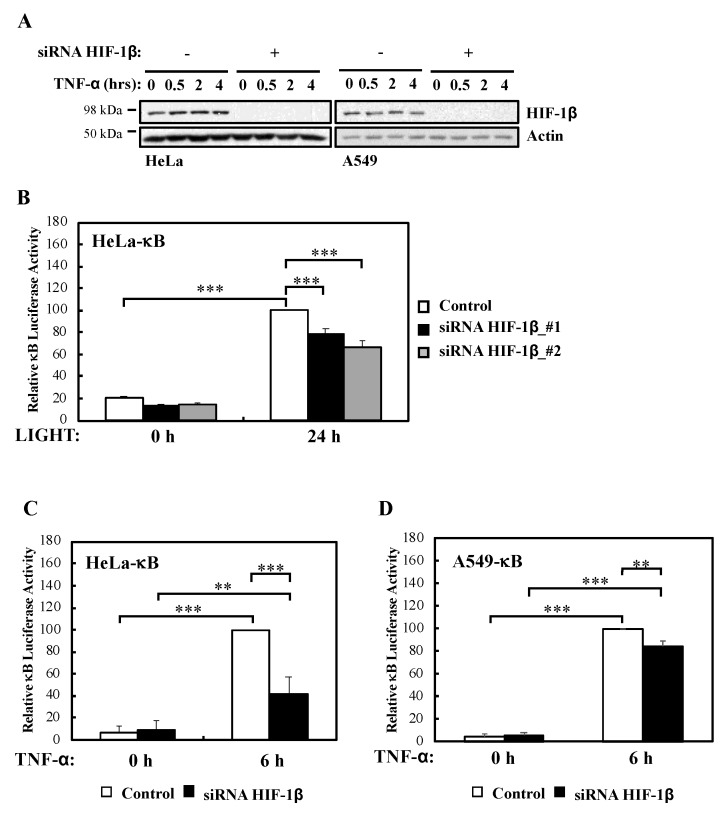
HIF-1β is required for full NF-κB reporter gene activity following non-canonical and canonical stimuli. (**A**) HeLa and A549 cells were transfected with control or HIF-1β siRNA oligonucleotides prior to treatment with 10 ng/mL TNF-α for the indicated periods of time before lysis and Western blot analysis. Actin was used as a loading control. (**B**) HeLa-κB luciferase cells were transfected with control and HIF-1β siRNAs for 24 h prior to treatment with 100 ng/mL LIGHT for another 24 h prior to lysis and luciferase activity measurements. All values were normalised to untreated sample. Graph depicts mean and SEM from four independent biological experiments. One way ANOVA analysis was performed and levels of significance was determined as follows: *** *p* < 0.001. (**C**) HeLa-κB luciferase cells were transfected as in *A*, but 10 ng/mL TNF-α was added for 6 h prior to lysis and luciferase measurements. Graph depicts mean and SEM from four independent biological experiments. One way ANOVA analysis was performed and levels of significance are indicated as follows: ** *p* < 0.01, *** *p* < 0.001. (**D**) A549- κB luciferase cells were treated as in *1C*. Graph depicts mean and SEM from three independent biological experiments. One way ANOVA analysis was performed and levels of significance determined as follows: ** *p* < 0.01, *** *p* < 0.001.

**Figure 2 ijms-21-03000-f002:**
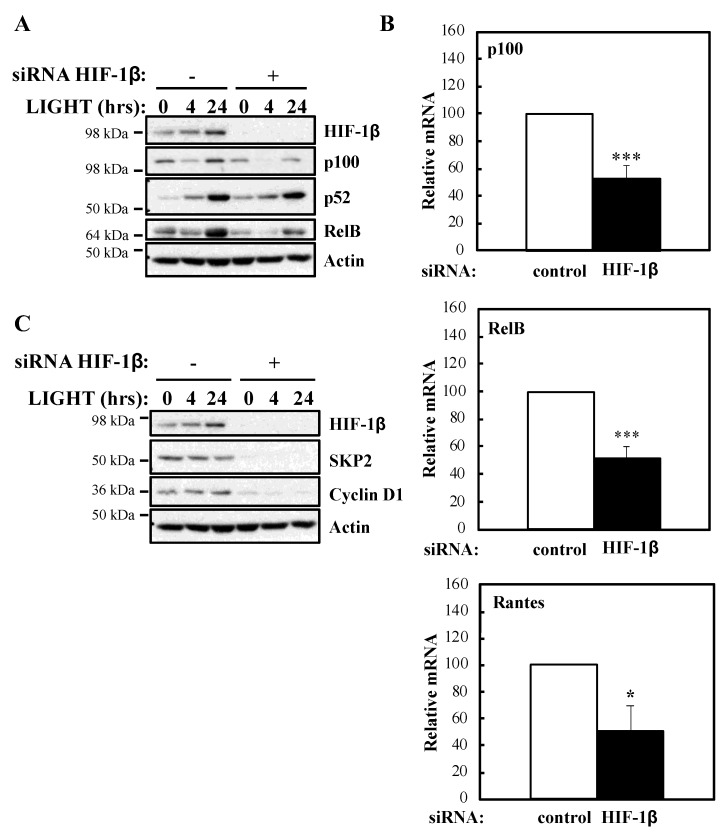
HIF-1β is required for full NF-κB activity in response to LIGHT. (**A**) HeLa cells were transfected with control or HIF-1β siRNA oligonucleotides. Where indicated, cells were treated with 100 ng/mL LIGHT for 4 or 24 h prior to whole cell lysis and Western blot analysis. Actin was used as a loading control. (**B**) HeLa cells were transfected with control or HIF-1β siRNA oligonucleotides for 24 h prior to treatment with 100 ng/mL LIGHT for other 24 h prior to lysis and RNA extraction. Following cDNA synthesis, qPCR was performed using primers for the indicated targets. Graphs depict mean and SEM from three independent biological experiments. Student *t*-test analysis was performed for each gene and levels of significance determined as: * *p* < 0.05, *** *p* < 0.001. (**C**) HeLa cells were treated and analysed as in *2A*.

**Figure 3 ijms-21-03000-f003:**
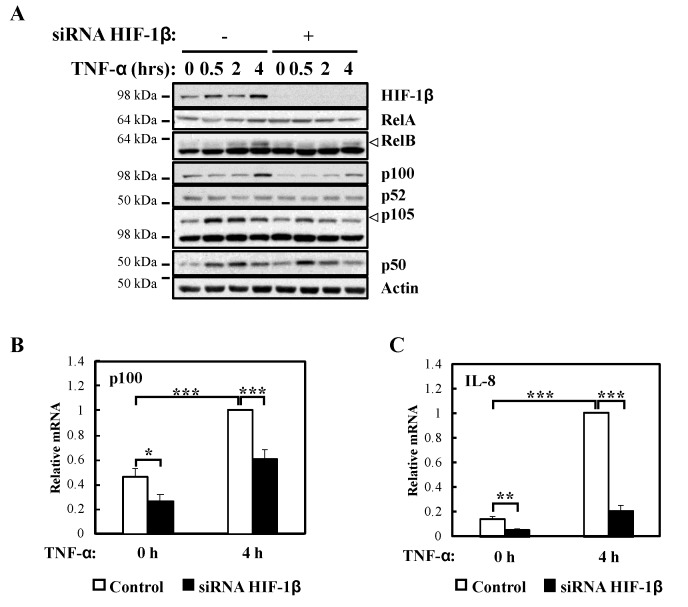
HIF-1β is required for full NF-κB activity in response to TNF-α. (**A**) HeLa cells were transfected with control or HIF-1β siRNA oligonucleotides for 48 h. Where indicated, cells were treated with 10 ng/mL TNF-α for 0.5, 2 or 4 h prior to lysis and Western blot analysis. Actin was used as a loading control. (**B**) HeLa cells were transfected as in *3A*, but also treated where indicated with 10 ng/mL TNF-α for 4 h prior to lysis and RNA extraction. Following cDNA synthesis, qPCR was performed using p100 primers. Graphs depict mean and SEM from four independent biological experiments. Student *t*-test analysis was performed and levels of significance determined as: * *p* < 0.05, *** *p* < 0.001. (**C**) HeLa cells were transfected as in *3A*, but also treated where indicated with 10 ng/mL TNF-α for 4 h prior to lysis and RNA extraction. Following cDNA synthesis, qPCR was performed using IL-8 primers. Graph depicts mean and SEM from three independent biological experiments. One way ANOVA analysis was performed and levels of significance indicated as follows: ** *p* < 0.01, *** *p* < 0.001.

**Figure 4 ijms-21-03000-f004:**
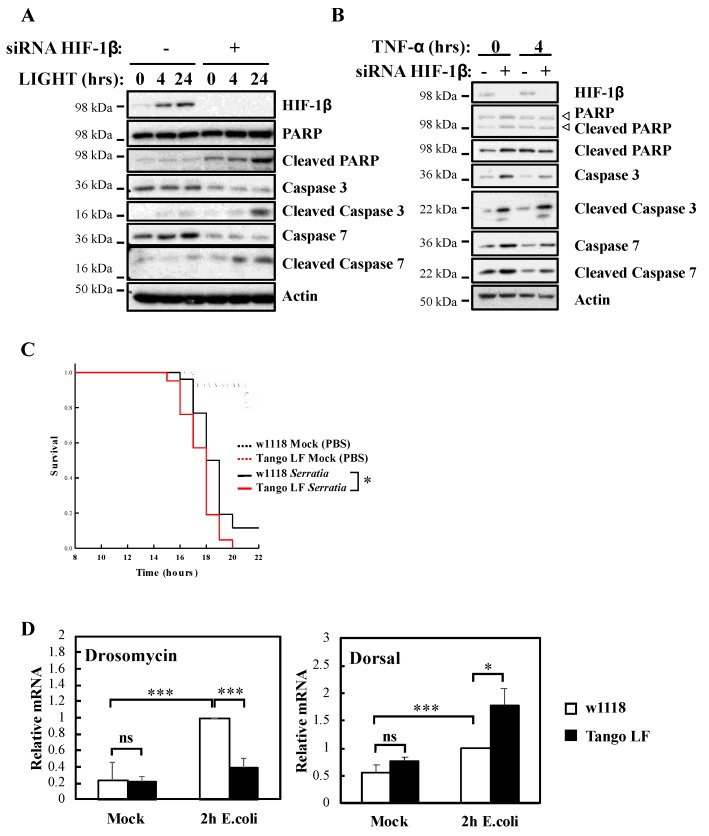
HIF-1β is required for viability of cells and *Drosophila melanogaster* following infection. (**A**) HeLa cells were transfected with control or HIF-1β siRNA oligonucleotides. Where indicated, cells were treated with 100 ng/mL LIGHT for 4 or 24 h prior to whole cell lysis and Western blot analysis. Actin was used as a loading control. (**B**) HeLa cells were transfected as in *4A*, but treated with 10 n/mL TNF-α for 4 h prior to lysis and Western blot analysis. Actin was used as a loading control. (**C**) Wild-type adult flies (*w1118*) and HIF-1β (*Tango*) loss-of-function flies (Tango LF) were pricked using a thin needle dipped in a diluted overnight culture of *Serratia marcescens* Db10 (OD_600_ = 0.2) or in a saline solution (PBS, Mock). Groups of 100 flies were used and kept at room temperature. Survival was monitored and expressed as the ‘estimated probability of survival’. The *p*-value was obtained from log-rank statistical analysis: * *p* < 0.05. (**D**) Wild-type adult flies (*w1118*) and HIF-1β (*Tango*) loss-of-function flies (Tango LF) were pricked using a thin needle dipped in a culture of *Escherichia coli* or in a saline solution (PBS, Mock). RNA was extracted 2 h later. Following cDNA synthesis, qPCR was performed using the indicated primers. Graphs depict mean and SEM from three independent biological experiments. Student *t*-test analysis was performed for each gene and levels of significance determined as: ns = not significant, * *p* < 0.05; *** *p* < 0.001.

**Figure 5 ijms-21-03000-f005:**
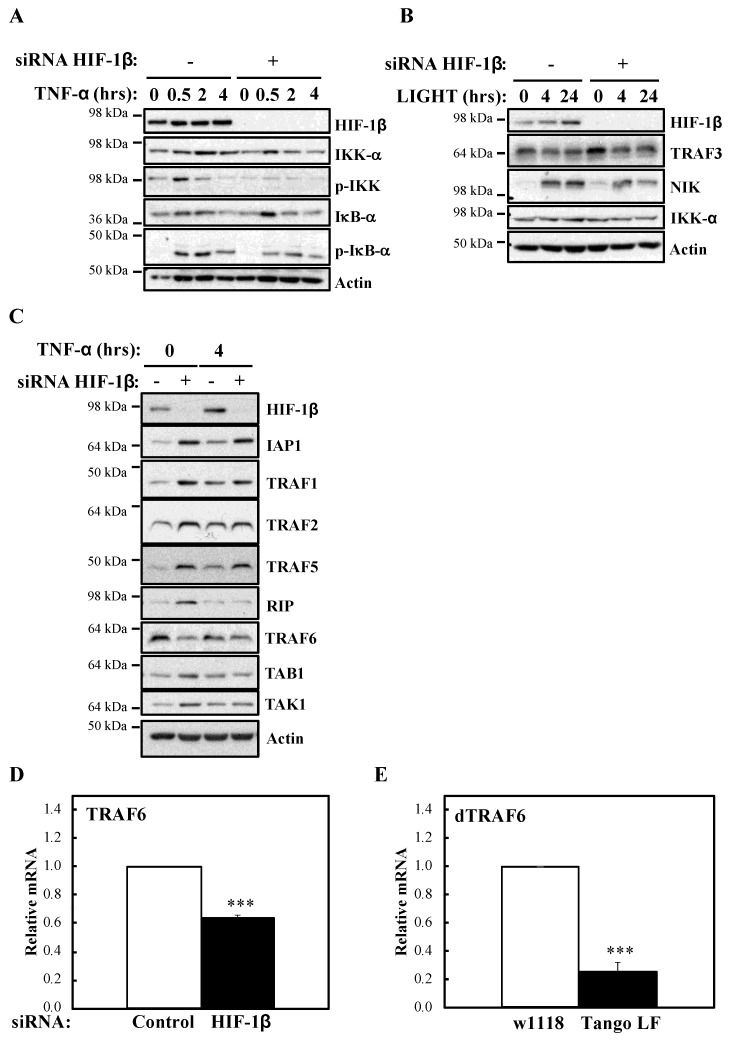
HIF-1β is required to control TRAF6 levels. (**A**) HeLa cells were transfected with control or HIF-1β siRNA oligonucleotides. Where indicated, cells where treated with 10 ng/mL TNF-α for the periods of time depicted prior to whole cell lysis and Western blot analysis. Actin was used as a loading control. (**B**) HeLa cells were transfected as in *5A*, but where indicated, cells were treated with 100 ng/mL LIGHT prior to whole cell lysis and Western blot analysis. Actin was used as a loading control. (**C**) HeLa cells were transfected as in *5A*, but treated with 10 ng/mL TNF-α for 4 h prior to cell lysis and Western blot analysis. Actin was used as a loading control. (**D**) HeLa cells were transfected with control or HIF-1β siRNA oligonucleotides for 48 h prior to lysis and RNA extraction. Following cDNA synthesis, qPCR was performed using specific primers for TRAF6. Graphs depict mean and SEM from three independent biological experiments. Student *t*-test analysis was performed and levels of significance determined as follows: *** *p* < 0.001. (**E**) Wild-type adult flies (*w1118*) and HIF-1β loss-of-function flies (*Tango LF*) were used for RNA extraction and cDNA synthesis. qPCR analysis was performed for the levels of *dTRAF6*. Graph depicts mean and SEM from three independent biological experiments. Student *t*-test analysis was performed and levels of significance as follows: *** *p* < 0.001.

**Figure 6 ijms-21-03000-f006:**
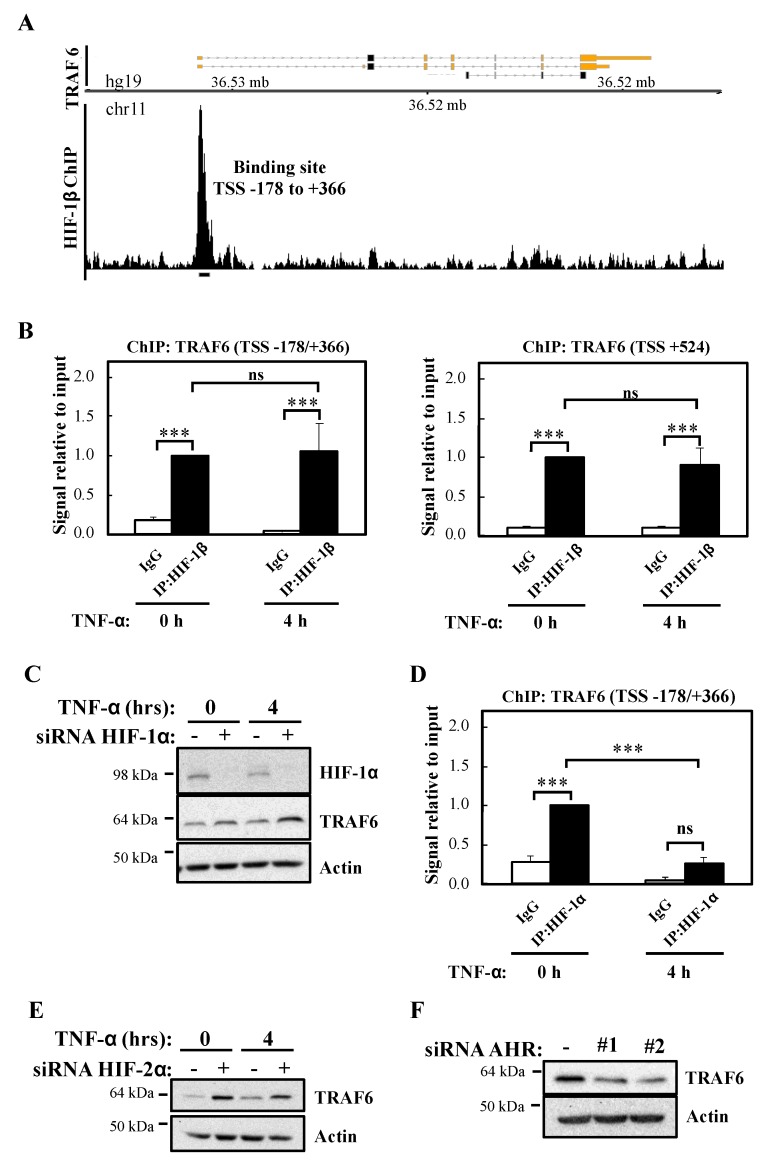
HIF-1β binds to TRAF6 gene and regulates its expression independently of HIF-1/2α. (**A**) Coverage tracks of HIF-1β ChIP-seq at the *TRAF6* gene in T47D cells. TSS = transcription start site. (**B**) ChIP for HIF-1β was performed in lysates derived from HeLa cells treated or not with 10 ng/mL TNF-α for 4 h prior to cross-linking and lysis. Occupancy at the indicated sites was analysed by qPCR. In all cases, Rabbit IgG was used as antibody control. Graphs depict mean and SEM from three independent biological experiments. One way ANOVA analysis was performed and levels of significance indicated as follows: ns = not significant, *** *p* < 0.001. (**C**) HeLa cells were transfected with control or HIF-1α siRNA oligonucleotide, but also, where indicated, treated with 10 ng/mL TNF-α for 4 h prior to lysis and Western blot analysis. Actin was used as a loading control. (**D**) ChIP for HIF-1α was performed in lysates derived from HeLa cells treated or not with 10 ng/mL TNF-α for 4 h prior cross-linking and lysis. Occupancy at the indicated site was analysed by qPCR. Rabbit IgG was used as antibody control. Graph depicts mean and SEM from three independent biological experiments. One way ANOVA analysis was performed and levels of significance determined as follows: ns = not significant, *** *p* < 0.001. (**E**) HeLa cells were transfected with control or HIF-2α siRNA oligonucleotides, but also, where indicated, treated with 10 ng/mL TNF-α for 4 h prior to lysis and Western blot analysis. Actin was used as a loading control. (**F**) HeLa cells were transfected with control or AHR siRNA oligonucleotides prior to lysis and Western blot analysis. Actin was used as a loading control.

**Figure 7 ijms-21-03000-f007:**
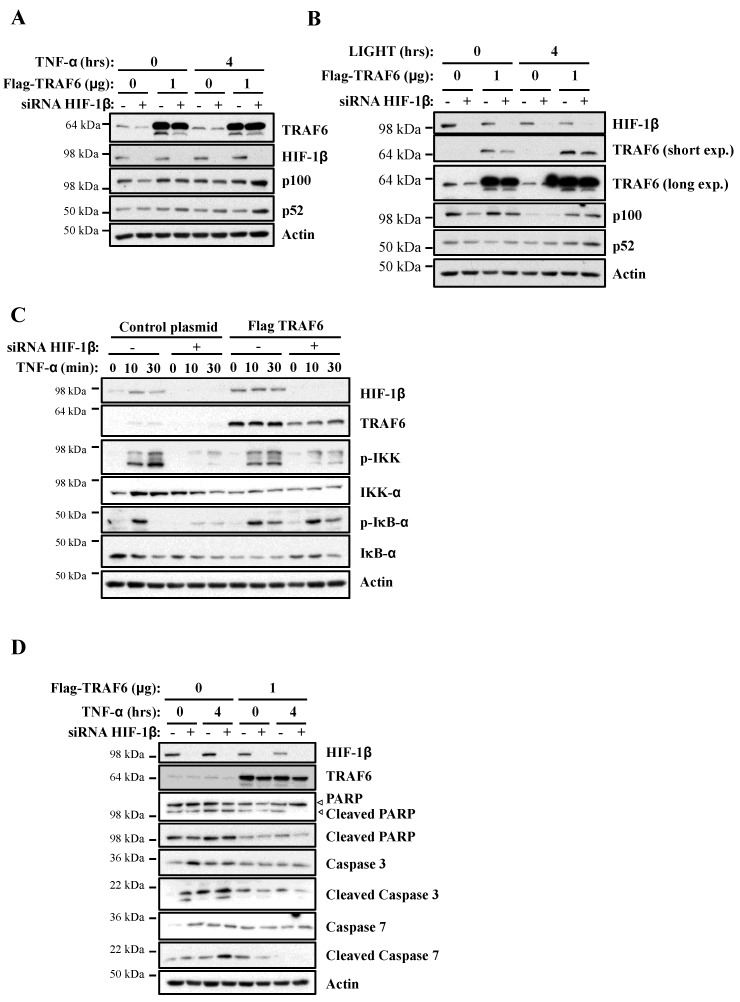
Exogenous TRAF6 is able to rescue HIF-1β loss in terms of NF-κB signalling and cell survival. (**A**) HeLa cells were transfected with control and HIF-1β siRNA oligonucleotides as well as with 1 μg of control or TRAF6 plasmid, prior to treatment with 10 ng/mL of TNF-α for 0 or 4 h. Whole cell lysates were collected, and Western blot analysis for the indicated proteins was performed. Actin was used as a loading control. (**B**) HeLa cells were transfected as in *7A*, but, where indicated, also treated with 100 g/mL LIGHT for 4 h prior to lysis and Western blot analysis. Actin was used as a loading control. (**C**) HeLa cells were transfected as in *7A*, and treated where indicated with 10 ng/mL TNF-α for the depicted periods of time prior to lysis and Western blot analysis. Actin was used as a loading control. (**D**) HeLa cells were transfected, treated and analyzed as in *7A*.

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
