# Peer review of "HIF-1β Positively Regulates NF-κB Activity via Direct Control of TRAF6"

_ijms, 2020, doi:10.3390/ijms21083000_

Round 1

Reviewer 1 Report

In this manuscript, D’Ignazio and colleagues investigate the role of HIF-1b, an essential component of the Hypoxia Inducible Factor (HIF) transcription factor, in non-canonical and canonical NF-kB signaling. By siRNA-mediated knockdown of HIF-1b in HeLa and A549 cell lines, the authors demonstrate that HIF-1b is necessary for full activation of the canonical and non-canonical NF-kB pathway upon TNFa and LIGHT stimulation, respectively. Depletion of HIF-1b is accompanied by upregulation of apoptotic markers (e.g. PARP, Caspase3) and signaling mediators (e.g. TRAF1/2/5, IAP1, RIP) as well as downregulation of transcription regulators (p100, RelB) and the mediator TRAF6. By ChIP HIF1b is found to associate with the TRAF6 promotor. Using overexpression of TRAF6, the authors argue that HIF1b affects canonical TNFa-induced NF-kB signaling pathway via inhibition of TRAf6 expression.

Despite the interesting observation that HIF1b is required for optimal NF-kB activation in response to TNFa, the work contains a significant conceptual flaw. Genetic work has clearly demonstrated that canonical NF-kB signaling in response to TNFa stimulation is independent of TRAF6, while IL1, CD40L or RANKL require TRAF6 (e.g. Lomaga et al., Cell 1999). Thus, the slight reduction in TRAF6 is certainly not able to explain the effects on TNF-induced NF-kB activation. If the authors want to make a conclusion, they need to knockdown TRAF6 and carefully determine what signaling pathways (e.g. RANKL, CD40, IL-1b) are affected. If residual NF-kB activation is observed, this activity should no longer be sensitive to HIF1b KD.

The authors suggest that HIF1b KD reduces TRAF6 in unstimulated cells. However, the same is actually true for p100 mRNA and protein levels as well as IL-8 mRNA and RelB protein amounts, which are all decreased in HIF1b KD cells even in the absence of stimulation. In contrast, protein expression of other TRAF1/2/5, IAP1, RIP and TAK1 is enhanced. Thus, HIF1b seems to be a transcription factor involved in the steady state regulation of many NF-kB family members and upstream regulators. These differences in multiple regulators may well explain the slight effects on NF-kB activation in response to different stimuli, because homeostatic control is disturbed. It seems unlikely that the effect is caused by the downregulation of a single protein like TRAF6.

Even though evolutionary conservation is interesting, the data on Drosophila melanogaster are not adding information with respect to the suggested mechanism in mammalian cells. The data on fly survival may be significant, but in how far the marginal difference in median survival (most likely < 1hour) is relevant is not clear. Moreover, these data are not linked to NF-kB signaling and it is quite unlikely that bacterial infection in Drosophila represents an in vivo model for TNFa stimulation in HeLa cells. Thus, I suggest to remove these data.

Specific points:

1) Experiments on TNFa activation and expression of many proteins (p100, TRAFs, IAP1 etc.) have all been performed with only one HIF1b siRNA and need to be repeated with a second siRNA. Better, HIF1b KO HeLa cells should be generated and subsequently reconstituted with HIF1b to confirm that these often small effects are indeed caused by loss of HIF1b.

2) Does HIF1b KD affect HIF1a expression/stability?

3) For most inductions, luciferase activity or mRNA levels after stimulation has been set to 100% (e.g. Fig. 1A, B, C; Fig. 2C, D). This is unusual, because it does not allow to directly see the fold induction. If a normalization is required, unstimulated cells should be set to 1.

4) The authors write about Fig 3A: ‘…in absence of HIF-1b, we observed at certain time points reduced levels of IkBa, as expected, due to it being a NF-kB target.’ This is not clear and needs to be quantified and mRNA levels of NFKBIA need to be determined.

5) Does overexpression of HIF-1b induce TRAF6 expression?

6) The presentation of Figure 5A is unacceptable. Overexpression of TRAF6 leads to a strong and robust NF-kB reporter gene activation so the authors need to compare values of unstimulated cells and not just simply show effects after TNF stimulation. Most likely, TRAF6 is completely overriding TNFa stimulation. Why is the effect of HIF1b KD so much weaker when compared to Fig. 1B? Is this effect specific for TRAF6 or what happens to siHIF1b effect when you overexpress for instance constitutively active kinase IKKb or p65/RelA? As pointed out earlier, TRAF6 is not required for TNFa stimulation to NF-kB, which actually strongly suggests that the effect is not TRAF6-specific.

Author Response

In this manuscript, D’Ignazio and colleagues investigate the role of HIF-1b, an essential component of the Hypoxia Inducible Factor (HIF) transcription factor, in non-canonical and canonical NF-kB signaling. By siRNA-mediated knockdown of HIF-1b in HeLa and A549 cell lines, the authors demonstrate that HIF-1b is necessary for full activation of the canonical and non-canonical NF-kB pathway upon TNFa and LIGHT stimulation, respectively. Depletion of HIF-1b is accompanied by upregulation of apoptotic markers (e.g. PARP, Caspase3) and signaling mediators (e.g. TRAF1/2/5, IAP1, RIP) as well as downregulation of transcription regulators (p100, RelB) and the mediator TRAF6. By ChIP HIF1b is found to associate with the TRAF6 promotor. Using overexpression of TRAF6, the authors argue that HIF1b affects canonical TNFa-induced NF-kB signaling pathway via inhibition of TRAf6 expression.

Despite the interesting observation that HIF1b is required for optimal NF-kB activation in response to TNFa, the work contains a significant conceptual flaw. Genetic work has clearly demonstrated that canonical NF-kB signaling in response to TNFa stimulation is independent of TRAF6, while IL1, CD40L or RANKL require TRAF6 (e.g. Lomaga et al., Cell 1999). Thus, the slight reduction in TRAF6 is certainly not able to explain the effects on TNF-induced NF-kB activation. If the authors want to make a conclusion, they need to knockdown TRAF6 and carefully determine what signaling pathways (e.g. RANKL, CD40, IL-1b) are affected. If residual NF-kB activation is observed, this activity should no longer be sensitive to HIF1b KD.

While we agree that investigating other stimuli for NF-kappaB is an interesting aspect, it is well beyond the scope of this study. In addition, genetic KO do not account for immediate effect or compensation, so every case is simply a model. In response to some of these questions, we have now included our data with a non-canonical stimulus LIGHT. Our data indicate that HIF-1beta is required for this as well and that TRAF6 is involved, at least partially, in both types of responses.

The authors suggest that HIF1b KD reduces TRAF6 in unstimulated cells. However, the same is actually true for p100 mRNA and protein levels as well as IL-8 mRNA and RelB protein amounts, which are all decreased in HIF1b KD cells even in the absence of stimulation. In contrast, protein expression of other TRAF1/2/5, IAP1, RIP and TAK1 is enhanced. Thus, HIF1b seems to be a transcription factor involved in the steady state regulation of many NF-kB family members and upstream regulators. These differences in multiple regulators may well explain the slight effects on NF-kB activation in response to different stimuli, because homeostatic control is disturbed. It seems unlikely that the effect is caused by the downregulation of a single protein like TRAF6.

While we agree that HIF-1beta can control many aspects of NF-kappaB signalling both directly and indirectly, we can only refer to the ones we have analysed. We can state that functionally TRAF6 can rescue the majority of defects observed in the absence of HIF-1beta. We have now included a discussion point on the other observed effects of HIF-1beta depletion in our cells, including the reviewer suggestion that HIF-1beta seems to be a TF involved in steady state regulation of many NF-kappaB signalling components.

Even though evolutionary conservation is interesting, the data on Drosophila melanogaster are not adding information with respect to the suggested mechanism in mammalian cells. The data on fly survival may be significant, but in how far the marginal difference in median survival (most likely < 1hour) is relevant is not clear. Moreover, these data are not linked to NF-kB signaling and it is quite unlikely that bacterial infection in Drosophila represents an in vivo model for TNFa stimulation in HeLa cells. Thus, I suggest to remove these data.

We think this is an important aspect of our study and have now included mRNA data on the role of tango in the regulation of NF-kappaB in response to infection. We have previously demonstrated that indeed this is a good model for NF-kappaB signalling, closely related to human cell response, although more simplified.

Specific points:

1) Experiments on TNFa activation and expression of many proteins (p100, TRAFs, IAP1 etc.) have all been performed with only one HIF1b siRNA and need to be repeated with a second siRNA. Better, HIF1b KO HeLa cells should be generated and subsequently reconstituted with HIF1b to confirm that these often small effects are indeed caused by loss of HIF1b.

Unfortunately we cannot do these at present. We also have gain of function experiments which demonstrate the opposite effect.

2) Does HIF1b KD affect HIF1a expression/stability?

Although, we have not investigated this in this study, we have done an extensive study on this in our previous publication van Uden et al., 2011. Plos Genetics.

3) For most inductions, luciferase activity or mRNA levels after stimulation has been set to 100% (e.g. Fig. 1A, B, C; Fig. 2C, D). This is unusual, because it does not allow to directly see the fold induction. If a normalization is required, unstimulated cells should be set to 1.

We have conducted this normalisation to reduce differences in the amplitude of the response.

4) The authors write about Fig 3A: ‘…in absence of HIF-1b, we observed at certain time points reduced levels of IkBa, as expected, due to it being a NF-kB target.’ This is not clear and needs to be quantified and mRNA levels of NFKBIA need to be determined.

We agree with the reviewer this is a complicated aspect, as IkappaB-alpha is controlled by mRNA and protein degradation. We could reliably quantify a reduction at basal level of IkB-alpha. We have correct this in the text.

5) Does overexpression of HIF-1b induce TRAF6 expression?

Unfortunately we have not investigated this, and at present we cannot perform this experiment.

6) The presentation of Figure 5A is unacceptable. Overexpression of TRAF6 leads to a strong and robust NF-kB reporter gene activation so the authors need to compare values of unstimulated cells and not just simply show effects after TNF stimulation. Most likely, TRAF6 is completely overriding TNFa stimulation. Why is the effect of HIF1b KD so much weaker when compared to Fig. 1B? Is this effect specific for TRAF6 or what happens to siHIF1b effect when you overexpress for instance constitutively active kinase IKKb or p65/RelA? As pointed out earlier, TRAF6 is not required for TNFa stimulation to NF-kB, which actually strongly suggests that the effect is not TRAF6-specific.

In our system, TRAF6 does not lead to a strong activation of the reporter. Data now presented in Sup Fig 6. We cannot compare this to IKKbeta or RelA expression at present, but on previous work, TRAF6 effect seems to be different.

Reviewer 2 Report

Comments on manuscript IJMS-754647 by D'Ignazio et al.

"HIF-1β positively regulates hif activity via direct control of TRAF6"

In this manuscript, the authors show that ablation of HIF-1β interferes with canonical and non-canonical NF-κB signalling and apoptosis. While the authors focused on NF-kB activation by pro-inflammatory stimuli, many effects were actually already seen under non-stimulated conditions. The absence of HIF-1β leads to decreased levels of TRAF6, and TRAF6 overexpression reverses many of the observed effects following HIF-1β RNAi.

Major comments:

  1. Title: "NF-κB activity" and not "hif activity".
  2. 2: both p100 and p105 are decreased upon siHIF-1β, nevertheless the processed p52 and p50 remain similar in control and siHIF-1β cells. How do the authors explain this? Is there an increased processing of the NF-κB subunits upon HIF-1β depletion despite lower p-IKK levels?
  3. Since RelB is clearly reduced even though p52 levels are normal (Figs. 2A-B), a reduction in non-canonical NF-κB signalling is consistent with the reduction in luciferase activity upon non-canonical LIGHT stimulation (Fig. 1) in siHIF-1β cells. However, non-canonical NF-κB signalling is largely ignored in the manusctipt.
  4. HIF-1β depletion induces apoptotic markers even in absence of TNFα (Fig. 2E), which also can be corrected by TRAF6 overexpression, suggesting an important role of HIF-1β and TRAF6 outside of the canonical NF-κB signalling.
  5. Canonical signalling is also reduced in the reporter gene assay (Fig. 1) but cannot be explained by presented NF-κB subunit levels because RelA and p50 appear equal (Fig. 2B). However, RelA and p50 are crucial for TNFα induced signalling. Did the authors consider alternative mechanisms? For instance, is it possible that the observed reduction in signalling is caused by any modification/phosphorylation of RelA? Are TBK1 levels influenced by siHIF-1β?
  6. 3: the statement about "specificity for TRAF6" on page 5 line 36 is confusing because also TRAFs1, 2 and 5 were regulated by HIF-1β RNAi, just in the other direction, whether or not TNFα was present. Similarly, TAB1 and TAK1 levels are increased in HIF-1β depleted cells in the absence of TNFα. The interpretation/discussion of these results must not ignore these important effects.
  7. No evidence for an involvement of NF-κB in the increased mortality in Tango flies is provided. Showing mRNA or protein levels of classical NF-κB targets would strengthen the statement that HIF-1β ablation in drosophila decreases TNFα signalling.
  8. The ablation of HIF-1β leads to decreased canonical and non-canonical signalling. Is TRAF6 also able to modulate the non-canonical pathway, especially since many observed phenotypes seem independent of TNFα. It has been reported that HIF-1β can bind RelB and alter, could this contribute to the observed effects?
  9. 6: what is a "HIF-1β binding site"? It is generally thought that HIF-1β/ARNT binds DNA only as heterodimer and HIF-1β monodimers have been discussed controversially. Similarly, AhR binds DNA only as heterodimer with ARNT. What is the binding partner of HIF-1β? The described phenotypes appear at least partially HIF-1α independent. The effect of the other known (conditional!) interactors of HIF-1β, HIF-2/3α and AhR, have not been explored. Especially, AhR has already been shown to interfere with NF-κB signalling, both inhibiting RelA:p50 mediated effects as well as binding RelB. With removal of one of AhR’s classical interactor (HIF-1β), interference with the other pathway might be enhanced. Hence, the contribution of AhR in observed phenotypes should not be neglected.

Minor comments:

  1. Introduction and p.6: ARNT is the "Aryl Hydrocarbon Receptor Nuclear Translocator" and not the "Aryl Hydrocarbon Nuclear Translocator"
  2. 1A: the error bars are amazingly low for this kind of experiment. Statistical tests to analyze the difference in the variance must not be performed against the normalizer (100%) which by definition has no variance. Fig. 4: same comment; the normalizer here is 1.0 without any error.
  3. 6: the HRE is the "Hypoxia Response Element" and not the "Hypoxia Responsible Element".
  4. S6: what is the p-value and why is it "0" in some instances?

Author Response

In this manuscript, the authors show that ablation of HIF-1β interferes with canonical and non-canonical NF-κB signalling and apoptosis. While the authors focused on NF-kB activation by pro-inflammatory stimuli, many effects were actually already seen under non-stimulated conditions. The absence of HIF-1β leads to decreased levels of TRAF6, and TRAF6 overexpression reverses many of the observed effects following HIF-1β RNAi.

Major comments:

  1. Title: "NF-κB activity" and not "hif activity".

This was a problem in the manuscript conversion and it is corrected now.

  1. 2: both p100 and p105 are decreased upon siHIF-1β, nevertheless the processed p52 and p50 remain similar in control and siHIF-1β cells. How do the authors explain this? Is there an increased processing of the NF-κB subunits upon HIF-1β depletion despite lower p-IKK levels?

Unfortunately, we are not able to investigate if HIF-1beta depletion increases or decreases p105 and/or p100 processing. Our results are best explained by possibly reduced levels of p65/p50 nuclear activity, which can be explained by reduced p-IKK and p-IkappaB-alpha levels.

  1. Since RelB is clearly reduced even though p52 levels are normal (Figs. 2A-B), a reduction in non-canonical NF-κB signalling is consistent with the reduction in luciferase activity upon non-canonical LIGHT stimulation (Fig. 1) in siHIF-1β cells. However, non-canonical NF-κB signalling is largely ignored in the manusctipt.

Yes, we agree with the reviewer and we have investigated non-canonical signalling more extensively in the absence of HIF-1beta. However, we did not include these data in the initial submission for simplicity and also due to the fact that HIF-1beta has already been shown to bind RelB and control its activity upon CD30 stimulation. However, we are now including our new data on non-canonical pathway, that confirms a generalised role for HIF-1beta in the control of NF-kappaB, which acts at least in part via TRAF6. We now show: 1) the reduced expression of p100 and RelB at mRNA level upon HIF-1beta knock-down and LIGHT treatment (in HeLa), in support of the results showed for the protein levels; 2) the reduced expression of target genes activated by the non-canonical pathway after HIF-1beta depletion as well as the induction of the apoptotic pathway also upon LIGHT stimulation and HIF-1beta depletion; 3) the reduced expression of the non-canonical NF-kappaB mediator NIK after HIF-1beta knock-down; 4) the restoration of the non-canonical NF-kappaB activity to normal levels in HIF-1β-depleted cells, following exogenous TRAF6 expression.

  1. HIF-1β depletion induces apoptotic markers even in absence of TNFα (Fig. 2E), which also can be corrected by TRAF6 overexpression, suggesting an important role of HIF-1β and TRAF6 outside of the canonical NF-κB signalling.

We thank the reviewer for this comment and completely agree, we have now included sentences in our results and discussion to indicate that this effect occurs even in the absence of stimulation.

  1. Canonical signalling is also reduced in the reporter gene assay (Fig. 1) but cannot be explained by presented NF-κB subunit levels because RelA and p50 appear equal (Fig. 2B). However, RelA and p50 are crucial for TNFα induced signalling. Did the authors consider alternative mechanisms? For instance, is it possible that the observed reduction in signalling is caused by any modification/phosphorylation of RelA? Are TBK1 levels influenced by siHIF-1β?

Unfortunately we did not investigate p-RelA levels or TBK1 levels in this study. These would be interesting experiments to conduct in the future. Our hypothesis is that RelA/p50 nuclear translocation and/or DNA binding is reduced in the absence of HIF-1beta. This is in accordance to reduced levels of p-IKK and p-IkappaB-alpha.

3: the statement about "specificity for TRAF6" on page 5 line 36 is confusing because also TRAFs1, 2 and 5 were regulated by HIF-1β RNAi, just in the other direction, whether or not TNFα was present. Similarly, TAB1 and TAK1 levels are increased in HIF-1β depleted cells in the absence of TNFα. The interpretation/discussion of these results must not ignore these important effects.

We have now corrected this aspect and changed our results and discussion accordingly.

4.No evidence for an involvement of NF-κB in the increased mortality in Tango flies is provided. Showing mRNA or protein levels of classical NF-κB targets would strengthen the statement that HIF-1β ablation in drosophila decreases TNFα signalling.

We thank the reviewer for this comments. We have included mRNA levels of NF-kappaB targets following E.coli infection. While levels of Drosomycin are reduced in the absence of tango, other targets are elevated. This could be due to loss of HIF-1alpha function, Sima, which we have demonstrated is required for efficient control of NF-kappaB activation in flies (Bandarra et al., 2015 DMM). We have include this in our discussion.

5.The ablation of HIF-1β leads to decreased canonical and non-canonical signalling. Is TRAF6 also able to modulate the non-canonical pathway, especially since many observed phenotypes seem independent of TNFα. It has been reported that HIF-1β can bind RelB and alter, could this contribute to the observed effects?

Our results indicate that TRAF6 can also modulate the response to non-canonical stimulation as well. So we hypothesise that HIF-1beta can regulate non-canonical pathway by at least 2 different mechanisms, via TRAF6 and by RelB binding, which has been published before.

6: what is a "HIF-1β binding site"? It is generally thought that HIF-1β/ARNT binds DNA only as heterodimer and HIF-1β monodimers have been discussed controversially.

The reviewer is correct. We have changed the terms used. Our analysis was conducted searching for putative HRE1 (5’-GCGTG-3’) and HRE2 (5’-ACGTG-3’), hypothesizing HIF-1beta was bound as a dimer to a HIF-alpha subunit, but we also considered the putative AHR:ARNT binding sites identified as 5’-CCACGCTTCC-3’, 5’-ACACGCGCGT-3’, and 5’-ACGCGCGTGT-3’ by ALGGEN-PROMO.

7: Similarly, AhR binds DNA only as heterodimer with ARNT. What is the binding partner of HIF-1β? The described phenotypes appear at least partially HIF-1α independent. The effect of the other known (conditional!) interactors of HIF-1β, HIF-2/3α and AhR, have not been explored. Especially, AhR has already been shown to interfere with NF-κB signalling, both inhibiting RelA:p50 mediated effects as well as binding RelB. With removal of one of AhR’s classical interactor (HIF-1β), interference with the other pathway might be enhanced. Hence, the contribution of AhR in observed phenotypes should not be neglected.

Indeed, this is an excellent question. We were able to rule out HIF-1alpha and HIF-2alpha. We have now included data for HIF-2alpha in the manuscript. Our cells do not express HIF-3alpha. Using siRNA, our data suggests AHR is required for TRAF6 protein expression in unstimulated conditions, but not at mRNA level, suggesting an additional partner is required for HIF-1beta regulation of TRAF6. We have include this in our discussion. 

Minor comments:

  1. Introduction and p.6: ARNT is the "Aryl Hydrocarbon Receptor Nuclear Translocator" and not the "Aryl Hydrocarbon Nuclear Translocator"

We apologise for this mistake and have now corrected it.

2: 1A: the error bars are amazingly low for this kind of experiment. Statistical tests to analyze the difference in the variance must not be performed against the normalizer (100%) which by definition has no variance. Fig. 4: same comment; the normalizer here is 1.0 without any error.

We have normalised it to the stimulated samples due to differences in the amplitude of response. The variance can be appreciated when looking at the non-stimulated levels. In any case, the comparisons always include all conditions, so the net effect is similar but make the representation easier to follow.

3: the HRE is the "Hypoxia Response Element" and not the "Hypoxia Responsible Element".

We thank the reviewer for pointing this mistake and we have now correct it.

4:S6: what is the p-value and why is it "0" in some instances?

This is due to the fact that the GEPIA 2.0 software rounds up the p value  to zero when it goes below a certain numbers of magnitude.

Reviewer 3 Report

There is general interest in the understanding of inflammation and hypoxia, both in normal immunological function and in cancer. Here the authors show that knockdown of Hif1b reduces NFkB activation and increases apoptotic markers in cancer cell lines. Hif1b knockdown leads to reduced TRAF6 expression, and the authors show evidence that TRAF6 is a target gene of Hif1b in cancer cell lines and in drosophila. Finally, TRAF6 over-expression rescues the Hif1b knockdown phenotype. The authors also show that Tango loss of function in drosophila has a slight defect in survival when flies are challenged with Serratia marcescens, however the results are not entirely convincing. Overall, there are some interesting findings presented. The connection between Hif1b and TRAF6 is novel and potentially relevant to future studies in immunology and cancer. Studies were performed in HeLa and A549 cells as well as in drosophila, so the take-away message is a bit more general and not necessarily studied in depth within a particular tissue-type. Future studies could go deeper into a model system of organ development or tumor initiation to understand the functional relevance.

  1. In Figure 1, the authors should demonstrate efficient knockdown at the protein level in HeLa and A549 cells using western blot within the main figure.

  1. In Figure 2B, the RelB blot appears to have been excessively cropped and cannot be interpreted as it is shown.

  1. Please show a western blot confirming Hif1b over-expression related to Figure S2B.

  1. The writing on page 4 lines 2-3 refers to drosophila and Figure 2E-F. In the following lines (lines 4-6 on page 4) there is no reference to the use of HeLa cells. However, Fig 2E is performed in HeLa cells so it should be clearly stated, and not mixed in with drosophila.

  1. The results in figure 2F are not especially convincing of an important role for Tango. Please consider showing this graphically in other ways. Are there other measures of the immune response in drosophila that you could use to strengthen the statements made?

  1. In Figure 3A, the levels of IkB-a are not convincingly reduced in the Hif1b knockdown cells. The levels look higher in the knockdown cells after 0.5 hours of TNFa treatment. This makes it difficult to understand the significance of the statement “In the absence of HIF-1b, we observed at certain time points reduced levels of IkB”

Author Response

There is general interest in the understanding of inflammation and hypoxia, both in normal immunological function and in cancer. Here the authors show that knockdown of Hif1b reduces NFkB activation and increases apoptotic markers in cancer cell lines. Hif1b knockdown leads to reduced TRAF6 expression, and the authors show evidence that TRAF6 is a target gene of Hif1b in cancer cell lines and in drosophila. Finally, TRAF6 over-expression rescues the Hif1b knockdown phenotype. The authors also show that Tango loss of function in drosophila has a slight defect in survival when flies are challenged with Serratia marcescens, however the results are not entirely convincing. Overall, there are some interesting findings presented. The connection between Hif1b and TRAF6 is novel and potentially relevant to future studies in immunology and cancer. Studies were performed in HeLa and A549 cells as well as in drosophila, so the take-away message is a bit more general and not necessarily studied in depth within a particular tissue-type. Future studies could go deeper into a model system of organ development or tumor initiation to understand the functional relevance.

  1. In Figure 1, the authors should demonstrate efficient knockdown at the protein level in HeLa and A549 cells using western blot within the main figure.

We have now included these controls in the main figure as requested.

  1. In Figure 2B, the RelB blot appears to have been excessively cropped and cannot be interpreted as it is shown.

Our original RelB antibody produced an unspecific band close to the RelB band. We have now included the less cropped blot, indicating the specific band.

  1. Please show a western blot confirming Hif1b over-expression related to Figure S2B.

We have now included these as requested.

  1. The writing on page 4 lines 2-3 refers to drosophila and Figure 2E-F. In the following lines (lines 4-6 on page 4) there is no reference to the use of HeLa cells. However, Fig 2E is performed in HeLa cells so it should be clearly stated, and not mixed in with drosophila.

We apologise and have now corrected this problem.

  1. The results in figure 2F are not especially convincing of an important role for Tango. Please consider showing this graphically in other ways. Are there other measures of the immune response in drosophila that you could use to strengthen the statements made?

As requested by another reviewer we have included mRNA data for NF-kappaB targets in the absence of tango in response to infection. This demonstrates a complicated picture due to the indirect effect on loss of Sima (HIF-1alpha) function as well.

  1. In Figure 3A, the levels of IkB-a are not convincingly reduced in the Hif1b knockdown cells. The levels look higher in the knockdown cells after 0.5 hours of TNFa treatment. This makes it difficult to understand the significance of the statement “In the absence of HIF-1b, we observed at certain time points reduced levels of IkB”

We agree with the reviewer, as IkappaB-alpha are controlled both by mRNA and degradation pathway. We have now corrected this statement in our text.

Round 2

Reviewer 1 Report

In my view my main point and serious concern is not addressed. The slight reduction in TRAF6 expression after HIF1beta knock-down cannot explain the decline in TNFa-induced NF-kB activation, because TRAF6 is simply not involved in TNFa-induced canonical NF-kB signaling (Lomaga et al., 1999). TRAF6 is required for IL-1 or CD40-tiggered NF-kB activation, so the authors just looked at the wrong pathway. Without clear and convincing data that a reduction of TRAF6 by a mild knock-down is reducing TNFa signaling to NF-kB in the system analyzed, I do not see a relevance for the presented data on the HIF1b-TRAF6 axis for TNFa signaling.

Author Response

We are sorry that the reviewer has this opinion. We have clearly demonstrated an efefct for TRAF6 in all NF-kappaB signalling pathways we have analysed, non-stimulated, TNF-alpha and canonical. This is evident from the rescue experiments we provided.

Reviewer 2 Report

The experiments shown in Figures 3 B and C have been performed in a similar manner and hence must also be displayed in a similar manner. Normalize only to the untreated (0 h TNF-a or IL-8) control siRNA values; do not normalize pairwise.

By convention, open bars (no pattern required!) should be used for controls, filled bars for experiments. Apply to the entire manuscript.

Author Response

The experiments shown in Figures 3 B and C have been performed in a similar manner and hence must also be displayed in a similar manner. Normalize only to the untreated (0 h TNF-a or IL-8) control siRNA values; do not normalize pairwise.

We have now replaced this graph with the format the reviewer requested.

By convention, open bars (no pattern required!) should be used for controls, filled bars for experiments. Apply to the entire manuscript.

We have changed the colour scheme as suggested throughout the manuscript.

Reviewer 3 Report

Authors have responded to comments. I hope all authors can stay safe and healthy during this challenging period.

Author Response

We thank the reviwer for their kind words and we also hope that they safe and well!